# Overcoming Technical and Market Barriers to Enable Sustainable Large-Scale Production and Consumption of Insect Proteins in Europe: A SUSINCHAIN Perspective

**DOI:** 10.3390/insects13030281

**Published:** 2022-03-12

**Authors:** Teun Veldkamp, Nathan Meijer, Frank Alleweldt, David Deruytter, Leen Van Campenhout, Laura Gasco, Nanna Roos, Sergiy Smetana, Ana Fernandes, H. J. van der Fels-Klerx

**Affiliations:** 1Wageningen Livestock Research, P.O. Box 338, 6700 AH Wageningen, The Netherlands; 2Wageningen Food Safety Research, P.O. Box 230, 6700 AE Wageningen, The Netherlands; nathan.meijer@wur.nl (N.M.); ine.vanderfels@wur.nl (H.J.v.d.F.-K.); 3CIVIC Consulting GMBH, Potsdamer Strasse 150, 10783 Berlin, Germany; alleweldt@civic-consulting.de; 4INAGRO, Ieperseweg 87, 8800 Roselare, Belgium; david.deruytter@inagro.be; 5Research Group for Insect Production and Processing, Department of Microbial and Molecular Systems (M2S), Geel Campus, KU Leuven, Kleinhoefstraat 4, 2440 Geel, Belgium; leen.vancampenhout@kuleuven.be; 6Department of Agricultural, Forest, and Food Sciences, Università degli Studi di Torino, Largo P. Braccini 2, 10095 Grugliasco, Italy; laura.gasco@unito.it; 7Department of Nutrition, Exercise and Sports, University of Copenhagen, Rolighedsvej 26, 1958 Frederiksberg, Denmark; nro@nexs.ku.dk; 8German Institute of Food technologies (DIL e.V.), Prof. Von Klitzing Strasse 7, 49610 Quakenbrueck, Germany; s.smetana@dil-ev.de; 9Sociedade Portuguesa de Inovacao Consultadoria Empresarial e Fomento da Inovacao SA, Av Marechal Gomes da Costa, 1376 Porto, Portugal; anafernandes@spi.pt

**Keywords:** alternative proteins, edible insects, feed, food, insect rearing, novel proteins, opportunities, processing, safety, sustainability

## Abstract

**Simple Summary:**

Insects are increasingly being used in Europe as a new or alternative source of protein for both direct human consumption and ingredients for feed and food production. Upscaling edible insect production and processing to a sustainable industrial sector is critical to supply the market and meet the foreseen future demands. In a market where transition to more sustainable protein sources is one of the primary challenges, sustainable insect products can contribute to a circular and sustainable economy as well as food security. SUSINCHAIN (SUStainable INsect CHAIN) is a European Horizon 2020 project that aims to contribute to overcome technical and market barriers to enable sustainable large-scale production and consumption of insect proteins in Europe by generating and sharing knowledge, as well as testing, piloting, and demonstrating newly developed insect chain innovations and increasing societal engagement. This article provides an outline of the various obstacles to upscaling of the insect sector and the project’s contributions to overcome these. The project covers the topics of: market opportunities, consumer perception, optimization of insect rearing conditions and substrates, insect transportation and processing techniques, application of insect products in food and feed, food safety issues in insect production and processing, together with economic and environmental sustainability. The project’s outcomes will provide tools for scaling up and commercializing the European insect sector.

**Abstract:**

The expected global population growth to 9.7 billion people in 2050 and the significant change in global dietary patterns require an increase in global food production by about 60%. The protein supply for feed and food is most critical and requires an extension in protein sources. Edible insects can upgrade low-grade side streams of food production into high-quality protein, amino acids and vitamins in a very efficient way. Insects are considered to be the “missing link” in the food chain of a circular and sustainable economy. Insects and insect-derived products have entered the European market since first being acknowledged as a valuable protein source for feed and food production in around 2010. However, today, scaling up the insect value chain in Europe is progressing at a relatively slow pace. The mission of SUSINCHAIN (SUStainable INsect CHAIN)—a four-year project which has received funding from the European Commission—is to contribute to novel protein provision for feed and food in Europe by overcoming the remaining barriers for increasing the economic viability of the insect value chain and opening markets by combining forces in a comprehensive multi-actor consortium. The overall project objective is to test, pilot and demonstrate recently developed technologies, products and processes, to realize a shift up to Technology Readiness Level 6 or higher. In addition to these crucial activities, the project engages with stakeholders in the insect protein supply chain for feed and food by living labs and workshops. These actions provide the necessary knowledge and data for actors in the insect value chain to decrease the cost price of insect products, process insects more efficiently and market insect protein applications in animal feed and regular human diets that are safe and sustainable. This paves the way for further upscaling and commercialization of the European insect sector.

## 1. Introduction

According to the FAO, feeding a world population of 9.7 billion people in 2050 [1] will necessitate a 60 percent increase in overall food production from 2005/2007 to 2050 [2]. To effectively achieve such significant changes, an economy based on the circularity principle and taking resource constraints into account is required [3]. The European Commission (EC) has adopted an ambitious new Circular Economy Package to assist European businesses and consumers in making the transition to a stronger and more circular economy, in which resources are utilized in a more sustainable way [4]. This package includes waste-related legislation measures, with long-term goals of minimizing landfilling and increasing recycling and reuse. The package also contains an Action Plan to assist the circular economy in each step of the value chain, from production to consumption, repair and manufacturing, waste management, and secondary raw material feed-back, in order to close loops in product lifecycles. Insects offer a lot of potential for efficiently upgrading food waste streams and convert these into high value proteins. To help the European Commission to achieve its goal of reaching a circular economy, the scaling up of the highly efficient use of insects in the food chain will ensure more sustainable, robust supply chains with high consumer acceptability and appealing commercial potential.

Insect species have been proposed which are suited for on-farm large-scale production as a highly potential alternative source of protein for the feed and food markets in Europe. Insects can contribute to the food chain’s long-term viability by serving as the “missing link” that can resolve product lifecycle loops [5]. Farmed insect species can convert low-grade biomass into valuable feed and food ingredients. The relevant insect species (1) can be grown on organic side streams and have high conversion efficiency [5], (2) emit less greenhouse gases and ammonia than traditional livestock [6,7,8], (3) contain high-quality protein, amino acids, and vitamins for animal and human health [9,10], and (4) require fewer resources to produce the same amount of protein as traditional protein sources [7,11]. As a result, insects are increasingly being used as a new or alternative source of protein, either for direct human consumption or indirectly in reconstituted meals (containing insect-derived protein), as well as being a good protein source for petfood and feed. FAO supports the use of edible insects and encourages their inclusion in people’s diets all across the world [5]. The total turnover of insect feed operators is expected to exceed EUR 2 billion per year by the end of the decade [12]. In Europe, there is currently a demand–supply mismatch (or gap) between the various actors in the insect chain. If feed and food processors would like to employ insects as ingredients, they need a consistent quantity of a predetermined quality of insects (products). Currently, insect farms are unable to generate the requisite volumes at a constant quality. Insect protein products are currently not cost-competitive with established protein sources such as soybean meal. However, insects are becoming increasingly cost-effective among the other alternative proteins. By sharing knowledge and collecting data, as well as testing, piloting, and demonstrating newly developed innovations in the insect chain and increasing societal engagement, SUSINCHAIN (www.susinchain.eu, accessed on 13 February 2022) contributes to reducing the demand–supply mismatch and lowering the cost price of insect products.

Stakeholders in the insect value chain, both within and outside the consortium, have identified the necessary strategies and related activities to overcome the existing scaling challenges, as well as the most promising insect species. As a result, for feed and/or food production, the project focuses on Black Soldier Fly (BSF), Housefly (HF), Mealworm (MW), and Crickets (HC). In the next sections, the state-of-the-art on market opportunities, consumer acceptability, insect rearing techniques and optimizing and pre-treatment of substrates to grow insects, insect transport and processing, insect applications in feed and food, safety in the insect chain, and economic and environmental sustainability are presented and, thereafter, the contribution of the project is presented on these themes. The article concludes with the impact of the project on these industrial and societal challenges (Figure 1).

## 2. Project Contribution beyond the State-of-the-Art

### 2.1. Current Barriers, Market Opportunities, and Consumer Acceptability

Based on the results of a survey of insect chain operators conducted in the framework of SUSINCHAIN, and validated by a workshop with operators, the importance of specific barriers for upscaling from pilot to commercial production of insect proteins was assessed, covering barriers faced by insect chain operators in all four stages of the supply chain: rearing of insects, processing of insects, insects for feed and insects for food. Barriers affecting operations most related to the following issues:Production barriers: scale, technology, supply, data and guidance, as well as insect pests and diseases;Financial, price and market barriers: inputs/costs, demand/prices, access to finance and social acceptance of insect production/products;Safety-related barriers: lack of knowledge with respect to chemical and microbiological food safety hazards, and measures to control these, as well as issues related to allergenicity of workers and consumers;Regulatory barriers: legal restrictions and issues related to obtaining an operating license for insect production.

As described in the following subsections, several of the key barriers are addressed by activities under the SUSINCHAIN project. SUSINCHAIN only addresses research with substrates that are allowed according to European legislation. This means that biowaste sources such as manure, slaughter by-products, and catering waste were not included in the project as substrates.

Insects for food are currently often promoted as specialized, expensive foods such as exotic snack goods and ‘protein bars’. The marketing of insect-based food products has also been hampered by a lack of customer acceptance. More information on insect-based products might be offered, not only to raise knowledge of the positive, ethical, and health aspects of edible insects, but also to improve their overall image [13]. Many researchers study the role of labels in consumers’ purchase decisions [14,15,16] or the interaction between food labelling and consumer trust in choice experimental settings [17], and this existing research is complemented and extended by consumer research in the framework of SUSINCHAIN (see also below, Section 2.5). SUSINCHAIN investigates the potential of replacing animal proteins in everyday meals with insect proteins in a large-scale consumer experiment in two European countries in order to achieve an impact on making ordinary diets more sustainable. Additionally, a mixture of approaches is used to provide information regarding the role of several characteristics of customers’ trust in food choice decisions. Online choice experiments, concept maps, and an Information Display Matrix are used to determine the significance of attitudinal and trust-based issues. The project aims to boost consumer trust in insect-based foods, as well as insect-based food products from fish, poultry, and pigs fed insect proteins [18]. The project also aims to contribute to the upscaling of the European insect value chain (Figure 2) by facilitating the exchange of information, sharing best practices and past lessons learned. This is facilitated through the organization of stakeholder workshops, targeted at specific challenges. Additionally, the outcome of the project will support the development of policies for enabling and promoting large-scale commercialization of high-quality insect proteins for food and feed in the EU.

### 2.2. Optimization of Insect Rearing

The nutritional and physical (e.g., moisture content, temperature, and pH of the feeding substrate) value, as well as the requirements of BSF, HF, MW, and HC, should be considered when composing feed substrates for different insect species. These data, on the other hand, are not regularly compiled and applied in practice. Mechanical or biological pre-treatment of insect feed substrate may improve substrate digestibility. However, there is a scarcity of data on the impact of substrate pre-treatments. Insect colonies are susceptible to insect pathogen infection, which can result in increased insect mortality and malfunctions, as well as the death of the entire colony in some situations. Some of the major disease agents for three of the four insect species considered in the project (MW, HF, and HC) have already been identified. However, there is a scarcity of information on how to prevent the spread of these agents and attenuate their consequences. It is expected that future companies will specialize and focus on the reproduction of insects and their supply to other companies as the insect value chain is scaled up. To do so, difficulties concerning the storage and transportation of insect eggs and pupae must be overcome. For example, eggs are sensitive to temperature and humidity variations, which currently results in yield loss. Additionally, eggs can hatch during travel. If technologies for preserving eggs and delaying the hatching were accessible, the sector’s prospects would be expanded. Finally, the commercialization of by-products (e.g., frass or dead adults) is still difficult and therefore the insect value chain’s economic feasibility is limited.

SUSINCHAIN offers upscaling solutions for the primary insect sector by collecting and combining of available data on the nutritional and physical needs of insect species and store these in an accessible database. These are used to improve the feed substrate mixture and to validate this mixture through insect feeding trials. To maximize benefits in terms of circularity and sustainability, the focus is mostly on substrates that cannot be used directly by livestock. Following that, the feed substrate is further optimized by experimenting with various pre-treatment methods such as heat treatment, milling, and fermentation, and also, housing conditions are tested [19,20]. The research aims to create diagnostic protocols for detecting insect diseases in BSF, HF, MW, and HC rearing. It also shows how to use ecologically safe and targeted treatments to prevent and mitigate the most common insect pests and diseases [21]. To improve connectivity and specialization, the ideal and minimal transport conditions for insect eggs are studied. Finally, new opportunities are sought to expand the commercialization of insect rearing by-products [22,23].

### 2.3. Upscaling Insect Transport and Processing

With the expansion of the insect industry, the development of a practical logistic chain is required to enable stable storage and transportation of insects with little or no quality degradation. The evaluation of insect transport and processing procedures at an industrial scale is an unequivocal step in the development of the industry.

The availability of an efficient and low-cost drying method is a critical component of the logistic chain. Water transportation should, in fact, be avoided from an economic and shelf-life perspective. In Europe, most insect producers rely on the expensive and not very sustainable freeze- and oven-drying procedures. However, several technologies in the feed, food and biochemical industries exist at an industrial scale and have promising applications, but they have not been validated and demonstrated for insect matrices.

In SUSINCHAIN, the possibilities and limitations of applying a modified atmosphere, which is often used to store processed foods and to transport fruits and vegetables, are investigated for BSF for the first time [24]. The research related to drying in SUSINCHAIN focuses on drying procedures that are already accessible at industrial scale (TRL 5) but have yet to be used on a broad scale in Europe for insects, including microwave [25] and radio frequency drying. In addition, processing techniques are studied that do not include a drying stage (which benefits the insect production cost) and that are available on an industrial scale, but have never been used for insects. One such technology is High Moisture Extrusion. As demonstrated in SUSINCHAIN, the method can turn entire insects into a finished product that can widely be used in the food, feed, and petfood industries. The work in SUSINCHAIN related to processing also includes enzymatic treatment of insects followed by industrial centrifugation to separate insects into liquid and solid fractions, which can be applied as feed, food, or pet food ingredient, or the fractions can be subjected to biochemical purification.

### 2.4. Use of Insects as Feed

Many of the feed products on the market today do not contain processed animal proteins (PAPs), which are approved to be used in fish feed. Insect protein has similar properties to PAPs and is a good, sustainable substitute [26,27] and an excellent source of energy and digestible amino acids for livestock. Since insect proteins were approved for use in aqua feed, European insect farmers have commercialized about 1000 tonnes of insect protein. Currently, the aqua feed market uses around half of all insect-based animal feed produced in Europe, with that figure predicted to rise to 75% by 2030. In April 2021, the EU Member States voted positively on the authorization of insect PAPs in poultry and pig feed. This proposal represents a relevant milestone for the European insect sector, as it marks one of the key steps in the authorization process. In line with the EU procedures, the entry into force of this proposal took place on 7 September 2021 [28]. This brings up new commercial potential, and the impact of different processing methods on nutrient digestibility must be evaluated. Processing methods have a big impact on protein digestibility, so finding the right value for raw material digestibility is critical for feed formulation. Performance experiments have mostly been carried out at a laboratory or pilot scale so far, with high and non-commercial inclusion levels. An on-farm validation of the acquired data has not yet been performed, but it is urgently needed because it is well-recognized that the conditions (e.g., in terms of hygiene) and, therefore, the performance of the animals in small scale experiments may differ from those observed at industrial size. Preliminary research suggests that insects may play a beneficial function in the modification of animal gut microbiota, perhaps improving animal health [29,30,31] and acting as a “green antibiotic” solution, but results must be validated for on-farm conditions.

SUSINCHAIN focuses on the digestibility of insect protein in feed for fish, poultry and pigs, to deliver new updated and valuable information since the feed ban on PAPs has been lifted. In order to include the correct nutritional value in commercial feed composition optimization, the nutrient digestibility of insect protein is required. Because the nutrient digestibility of insect protein varies by animal species, the nutrient digestibility of insect protein in rainbow trout, seabass, salmon, broilers, and piglets is examined. The effects of insect inclusion in the feed on animal performance and health, as well as product quality, are then evaluated in large-scale trials. To conduct the large-scale studies, commercial diets containing two levels of insect meals (5 and 10%) are composed (TRL 6). The effects of bioactive substances (chitin, antimicrobial peptides, and lauric acid) provided by insect meal in diets on animal health are investigated utilizing hematological, biochemical, histological, and immunohistochemical investigations, as well as a metagenomics approach. The results are used to demonstrate that inclusion of insect protein in animal diets may improve animal performance, health, and product quality.

### 2.5. Use of Insects as Food

In general, the nutritional composition of edible insect species is similar to that of other animal-based meals. Since roughly 2013, a number of edible insects suited for farming have intermittently appeared in European markets, mostly as niche, exotic snack products such as entire dried flavored insects and various ‘protein bars’. The legislation for placing insects on the market as food in the EU member states is regulated by the Novel Food regulation. In 2018, the new Novel Food regulation (EU 2015/2283) replaced the original regulation (EC 258/97). The original regulation was interpreted differently in the member states, so insects were marketed as food in some countries and not in others. The new Novel Food regulation clearly defines edible insects as Novel Food. Insect products can be applied for approval to be placed on the market in all member states upon a scientific evaluation for safety by the European Food Safety Authority (EFSA). To date, products of MW (*Tenebrio molitor*), grasshopper (*Locusta migratoria*) and HC (*Acheta domesticus*) have been evaluated and approved as safe. Products of other species are currently under evaluation and expected to be approved to be released to the market. In the meantime, a transition regulation allows producers to keep products on the market following the original regulations. Despite the regulatory uncertainties, significant research and development efforts are being committed in creating technology for the cultivation and processing of insects as food ingredients [25], which might be used in processed foods such as pasta or bakery products [32,33]. Only in a few countries in Europe, such as Belgium [34,35], Finland [34,36] and the Netherlands [37], or in general Western countries [38], consumer acceptance of insect products was studied or consumers were exposed to such products. A 2015 survey in the Netherlands found that only a small percentage of customers had progressed from occasionally tasting insects to frequently eating them [39]. Even in societies with certain traditions of insect intake, barriers to consumption are multifaceted, with low familiarity and convenience being among the factors limiting future acceptance [40], demonstrating that barriers extend beyond novice consumers’ initial ‘disgust’ reaction.

The SUSINCHAIN project develops prototypes of insect-based food products suited for home dinner meal preparation in European families and consumer perception of their inclusion in a regular diet is assessed. Project partners had already developed certain insect-based ingredients and prototypes of processed food products made from mealworms and crickets [33,41]. Prior to consumer testing, the design and sensory quality of these insect-based products have gone through optimization to meet a set of quality criteria defined in the early stage of the project. The optimized products are tested for acceptance in ordinary dinner meals in potential early adopters of young, urban families in two European countries (Denmark and Portugal). Participating families will receive the products as a basket of products to be included in dinner meals three times a week over a six-week period, replacing the meat which would otherwise be the usual choice as the main protein source. The findings of this consumer intervention provide new and useful information to understand the barriers and motivations for consumers to incorporate insect products in their meals and to help insect growers and the food industry to establish consumer trust and identify possible pathways to larger markets [42]. Our ultimate goal is to transform insect-based foods from exotic snack foods to household kitchens, creating a viable alternative to traditional animal protein sources in the European food system.

### 2.6. Control Insect Safety

Evidence on the safety of using insects as feed and food has been gained in recent years, but knowledge gaps still persist. Potential food safety concerns in insects for food and feed, including the presence of microbiological and chemical hazards, allergenic compounds, and prions, were recently reviewed [43,44,45,46], highlighting current knowledge gaps. The main message was that safety should be assessed on a case-by-case basis, since different insect species behave in different ways when it comes to the transfer and accumulation of safety hazards from the substrate. In addition to the insect species, food safety is also influenced by the substrate itself, and by rearing and processing conditions. Cadmium and arsenic may be of concern in BSF and MW, respectively [47], whereas mycotoxins do not seem to accumulate in these species [48]. Importantly, insects appear to be able to metabolize mycotoxins [49,50], albeit additional research is needed to confirm this hypothesis. Other contaminants that may be of concern for reared insects are pesticides, medicines, hormones, dioxins and PCBs residues and more research is needed for these contaminants [43].

SUSINCHAIN takes a safety-by-design approach. Potential safety hazards of the insect species, substrate and rearing conditions are addressed in the project [45]. More specifically, the safety of the insects is investigated for the most promising rearing and processing techniques. For the substrates investigated in the project, potential hazards are prioritized. The prioritized hazards, including microbiological and chemical hazards, are then analyzed in the substrates. In addition, for several combinations of insect species and food safety hazards, transfer of the hazards to (larvae of) edible insects from substrates is investigated [51]. These include plant toxins in BSF and MW, and the pathogens *Salmonella* spp., *Staphylococcus aureus*, *Bacillus cereus s.l.*, and *Clostridium perfringens* in BSF and/or MW. The first studies on the transmission of pathogens to larvae of edible insects from the substrate have recently been published [22,51,52] and additional studies are currently ongoing. The capacity of insects to metabolize mycotoxin is also being studied in greater depth [53], covering several mycotoxins in BSF and MW. Black soldier fly larvae and MW (*Tenebrio molitor*) have been demonstrated to not store mycotoxins and to transfer them into their respective metabolites to some extent. In particular, aflatoxin contamination of crops is a global threat that has resulted in massive economic losses as well as major human and animal health issues. If insects can metabolize these highly toxic mycotoxins into non-toxic or substantially less toxic metabolites, insect rearing potential will be greatly enhanced, providing a boon for food security.

Frass can be used as a valuable co-material or fertilizer. However, more research into the nutritional value and safety of residual insect rearing material is needed [23]. Frass must be shown to be safe by establishing that it is free of environmental and food safety hazards such as heavy metals, *Salmonella* spp., and *E. coli*. Finally, to safeguard the production, processing, and use of insect products as food and feed, good safety practices for supply chain actors are also being developed.

### 2.7. Economic and Environmental Sustainability

Several European companies are currently operating large-scale insect production farms. When compared to traditional protein sources, advancements in the development of the European insect business are frequently associated with more favorable sustainability potentials of insects. Despite several studies on economic feasibility, societal acceptance, and environmental impact, the sector still faces a number of unanswered problems. Insects are considered as sources of proteins with fewer environmental impacts than meat products. Previous research indicated that their impact could be equivalent to that of chicken and pork products in terms of nitrous oxide emissions [7] and land use [54]. According to current research, some insect species have the potential to be used as a substitute for commercially available feed ingredients. Many studies have highlighted the possibility of replacing expensive protein sources of feed (fish meal and soybean meal) [55,56], which is specifically feasible due to the potential of agri-food waste, municipal waste [57] or manure use for insect feeding [58]. When it comes to renewability and digestibility, other authors point out that soyabean meal has a better exergy to energy transformation [59]. The majority of the findings are unusable by industry since they frequently report environmental impact on legally or technologically incorrect feeding substrates, or growing and processing technologies with lower TRL than what is currently available in industry. Systematization of potential feeds (non-utilized sources of biomass appropriate for insect feed) is currently a major challenge for the insect business, which can ensure both cost-effective production and environmental sustainability. There is presently no comprehensive database of available side streams that is applicable to the insect industry and usable for LCA study. In addition, an industrial scale is necessary for the development of particular industrial guidelines, product innovations, and a reliable comparison with traditional protein sources. The majority of current economic feasibility and environmental impact studies are conducted on a pilot or small industrial scale, with rates of 0.02–1 ton of insect biomass processed (dry weight basis) per day [8,60,61]. Insect production and processing at the industrial level for food and feed is also not taken into account. Separate production processes, incomplete data, different insect species, and only a few products are the focus of the studies. Direct industrial applications and credible advice for the selection of more environmentally friendly choices are not possible. The most significant obstacle to widespread adoption of sustainability principles in the insect rearing sector is a lack of sufficient data.

The Life Cycle Assessment (LCA) approach is used by SUSINCHAIN to provide viable answers and recommendations on environmental sustainability performance. However, its applicability is currently limited to a few situations and is not transferable to broader applications as general guidelines. To address data limitations in LCA practices for the insect industry, the project employs a modular modelling approach to cover the full range of insect production and processing parameters [62]. It is based on a Life Cycle Sustainability Assessment (LCSA) approach comprising several modules, each of which is responsible for a different aspect of insect biomass production and processing. To formulate a complete spectrum of present and emerging technologies, a matrix of insect production and processing modules based on data from the food and feed industry is updated with optimization algorithms to address the missing data points and uncertainty levels [63]. Insect, food, and feed sectors can use such a comprehensive database, which includes economic and environmental data, to identify optimal production–processing–consumption chains based on a few critical data points. The database is part of a Decision Support System that has a user-friendly interface and may be used by individual insect industry enterprises. As a result, the project will be able to industrialize the use of LCSA in the food and feed business, specifically in insect production.

## 3. Impact

The SUSINCHAIN project’s research and innovation activities have both short and long-term impacts. The project contributes to Europe’s efforts to achieve the Sustainable Development Goals (SDGs). The seventeen SDGs are all linked to feeding the world’s expanding population in a sustainable way, either directly or indirectly. The project specifically targets at least six of these SDGs (Table 1). The project’s contribution to the stated SDGs is inextricably linked to how we generate food for the world’s rising population, particularly the fundamental changes that are required for food production. Furthermore, in accordance with the COP 21 Paris Climate Agreement, which aims to strengthen the global response to the threat of climate change in the context of sustainable development and poverty eradication, project innovations help to slow the rise in global average temperature and foster low greenhouse gas emissions development without jeopardizing food production. While addressing climate change and natural resource limitations, the project contributes to ensuring sufficient, nutritious, safe, and affordable food for a rapidly growing global population.

Recent innovations are evaluated, piloted, and demonstrated at different stages of the insect value chain, including insect rearing, transportation and processing, and feed and food manufacturing, advancing them from TRL 5 to TRL 6 or higher. There are new market prospects for novel protein-based insect products for food (MW and HC) and feed (BSF and MW). Tools that promote consumer trust and acceptance of these novel food and feed products will result in increased acceptance of insect proteins by food chain actors. Insects convert low-grade organic waste streams into high-quality feed and food ingredients. Edible insects provide high-quality protein, amino acids, and vitamins to humans and livestock while emitting less greenhouse gases, ammonia, and water than traditional livestock and consuming fewer resources for the same amount of protein. When insects are added as the missing link to close product lifecycle loops, the food chain becomes more sustainable, resilient, and future-proof. Insect-based food items are developed and tested for nutritional and sensory qualities, as well as consumer acceptance in a regular diet, in large-scale studies. The novel insect-based food products are projected to have at least equivalent functionality and nutritional benefits to conventional food products, making them a viable option. Upscaling insect production has the potential to yield large amounts of safe and sustainable insect-based products of consistently high-quality protein for the feed and food chain, helping to meet future protein demand with a reduced environmental footprint than conventional protein.

Market opportunities and technical developments are selected based on their technical viability, and the expected impact is the empowerment of insect farmers and all other actors in the insect chain to create efficient, safe, and sustainable insect products. Innovations in insect rearing, transportation, processing, and applications in feed and food for the four insect species covered by the project increase the competitiveness of the insect value chain in Europe, but also globally, and offer new market opportunities. The project contributes to increase the ability of insect farming to supply high volumes of high-quality, safe, and sustainable goods, giving consumers more options for eating sustainably produced food.

One-third of all food is being wasted [72]. New food supply chains should result in healthier, more sustainable food production and consumption systems that generate less waste from one side and upcycle available waste for the preservation of nutrients in the food system. Solutions are provided for more environmentally friendly food production, as well as to persuade consumers to switch to more healthy and safe diets. Insect farming contributes to tackling societal challenges by feeding insects with co-products from the agri-food industries, mainly resources that are currently not used by, or no longer destined for, human consumption. Insect producers use co-products from the cereal, starch, fruit and vegetable supply chains or from local food processors (e.g., pastry and biscuits), local artisans (e.g., bakers), or even products from supermarkets that go unsold for technical or logistical reasons, and thus contribute to reduce waste. These products have low environmental footprints because they are co-products of the food chain and do not require any new processes for their use.

The project’s outcomes will have a positive impact on the environment and will help to mitigate the effects of climate change. Insect protein replaces traditional protein sources in feed and food, reducing reliance on traditional protein sources imported from outside the EU and the associated deforestation process in Latin America. Soybean meal is imported into the EU at a rate of 18.5 million MT (mega tonnes) each year. In the EU, replacing 10% of soybean meal with insect meal in animal diets would result in a reduction of 1.85 million MT soybean meal imports.

The project’s innovations enable insect protein production to be scaled up and commercialized. As a result of the upscaling of the insect chains for the four species in this project—BSF, HF, MW, and HC—more jobs will be created. Projections from the International Platform of Insects for Food and Feed [73] showed an expected increase in number of jobs in the European insect sector from ca. 1100 jobs in 2018 to over 100,000 jobs in 2030. The project contributes to these projections for the insect sector and helps to achieve these projections by providing knowledge, data and tools. Insects are classified as ‘farmed animals’, and the EU has placed restrictions on the feed that can be supplied to them. Insects intended for human food or animal feed are also subject to these limitations. Only plant-based products are allowed to be fed to insects. However, components of animal origin such as milk, eggs, and their products, honey, rendered fat, and blood products from non-ruminant animals are permitted. It is, however, prohibited to feed farmed animals other slaughtering or rendering-derived products, manure, or catering waste. If the product contains meat or fish, the same prohibition applies to unsold products from supermarkets or food industries (e.g., unsold products due to manufacturing or packaging defects). To completely close the loop and make the insect chain even more sustainable, further information and data on the safety of legally not yet approved substrates to grow insects is required.

## Figures and Tables

**Figure 1 insects-13-00281-f001:**
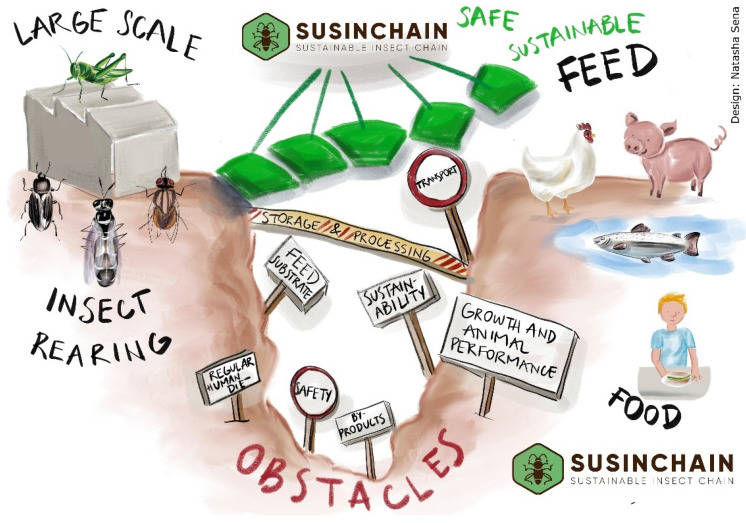
Challenges to be addressed to fill the gap between insect protein demand and supply.

**Figure 2 insects-13-00281-f002:**
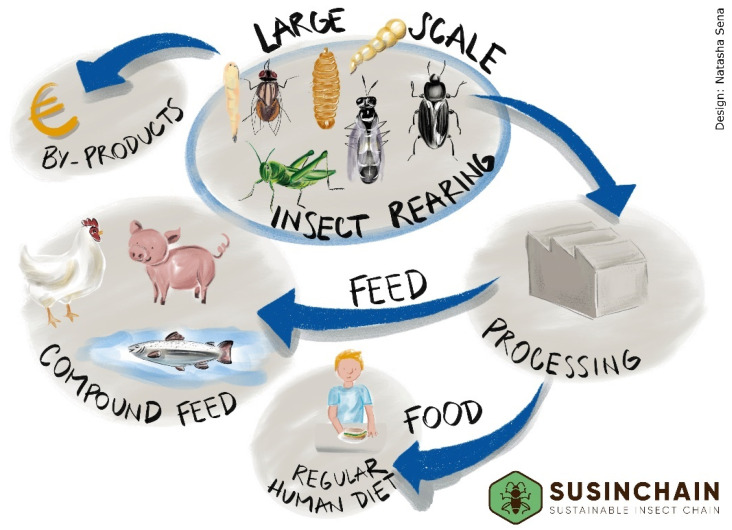
Insect production chain.

**Table 1 insects-13-00281-t001:** The contribution of the SUSINCHAIN project to the anticipated impacts.

Sustainable Development Goals	SUSINCHAIN Contribution
SDG2 ‘Zero hunger’	Provide new safe protein sources as there are 821 million being undernourished people worldwide in 2017 [64]
	Provide feed and food protein using less land
	Provide nutritious feed for poultry, pigs and fish, as well as human food
	Rearing on organic side streams to reduce organic waste
SDG3 ‘Good Health and Well-being’	Provide a feed source to ensure sufficient animal proteins for guaranteeing human well-being and health [65] because insects contain nutrients and dietary energy to meet the requirements of the human body as a part of a varied diet and for a rapidly growing world population
	Provide bioactive compounds (from insects) that have animal and human health benefits beyond simple nutritional values
	Bio-active compounds in insects may contribute to a reduced use of antibiotics in the livestock sector [66]
SDG9 ‘Industry, innovation and infrastructure’	Project innovations support the insect rearing, processing, and feed and food industry to progress from pilot to commercial production of insect proteins for food and feed
	Build a resilient infrastructure along the insect value chain, from industries with organic side streams up to food and feed industries selling insect protein-based products, to promote inclusive and sustainable industrialization and foster innovation
	Supports resource-efficient production systems, with value-chains based on new and more efficient use of organic side streams, residues and by-products, hereby contributing to maintaining and enhancing natural resources in Europe
SDG12 ‘Responsible Consumption and Production’	Insects can partially replace the use of conventional protein sources
	Organic side streams use, such as crop residues, contribute to a sustainable production pattern and a more circular economy
	The project’s insect based food production ensures close connection between dietary, environment and health aspects [67]
	The project contributes to future-proofing our food systems by making the systems more sustainable, resilient, responsible, diverse, competitive and inclusive
SDG13 ‘Climate Action’	Reduction in the contribution of livestock production to greenhouse gas production and climate change [68,69]
	Insect production induces lower greenhouse gas emission rates [7,70] than conventional livestock species [8], especially if consumed directly as food
	Project innovations contribute to climate change mitigation and, thus, contributes to meeting the COP 21 Paris Climate Agreement [71] which aims to strengthen the global response to the threat of climate change, in the context of sustainable development and efforts to eradicate poverty
SDG 15 ‘Life on Land’	SUSINCHAIN contributes to changes in livestock production that result in reduced use of land and reduced reduction in biodiversity

## Data Availability

No new data were created or analyzed in this study. Data sharing is not applicable to this article.

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
