# Peer review of "Overcoming Technical and Market Barriers to Enable Sustainable Large-Scale Production and Consumption of Insect Proteins in Europe: A SUSINCHAIN Perspective"

_insects, 2022, doi:10.3390/insects13030281_

Round 1

Reviewer 1 Report

An important and current topic, in the current mainstream research, Clearly written text, valuable and communicative graphics (diagrams, facilitate the assimilation of the content by the reader). abundant and up-to-date literature was used.

Specific comments

  1. What is the main question addressed by the research?

what are the challenges that must be overcome in order to bridge the gap between insect protein demand and supply

  1. Do you consider the topic original or relevant in the field, and if
    so, why?

The theme is original (in my op[inion), it contains a comprehensive approach to the description of production chains. Accurately describes and identifies barriers, including optimisation of insect rearing, upscaling insect transport and processing, use of insects as feed and food, control insect safety, economic and environmental sustainability

  1. What does it add to the subject area compared with other published
    material?

Completeness and versatility of factors taken into account, from production to safety

  1. What specific improvements could the authors consider regarding the
    methodology?

This is an Opinion Article, therefore there is no laboratory research methodology. These can be found in the cited works. It is hard for me to point to an improvement in the approach in analyzing the presented source publications.

  1. Are the conclusions consistent with the evidence and arguments
    presented and do they address the main question posed?

In my opinion – yes.

  1. Are the references appropriate?

Yes.

  1. Please include any additional comments on the tables and figures.

Figure 1 and 2 are transparent and convey the content well, synthetically. They are an example of a modern presentation of the most important thoughts and relationships between the processes described in the article. It is a modern use of visual skechnote

Table 1 is clear and shows the link between the goals of sustainable development well and the contribution of the SUSINCHAIN project to the anticipated impacts.

Author Response

Dear reviewer,

Thank you for your compliments on the manuscript and good to see that the manuscript is clear according to your answers to Specific comments 1 ... 7

Reviewer 2 Report

Just a few typos:

Line 106 - "Till" should be "Until"

Line 155 - Add "r" to "researches"

Line 306 - Remove "among"

Line 312 - This line might be missing a word or two between "alternative help". Double check for clarity.

Author Response

Dear Reviewer,

Thank you for your comments. With respect to your comments:

Just a few typos:

Line 106 - "Till" should be "Until" - Corrected

Line 155 - Add "r" to "researches" - Corrected

Line 306 - Remove "among" - Corrected

Line 312 - This line might be missing a word or two between "alternative help". Double check for clarity. - The sentence has been changed into: 'The findings of this consumer intervention provide new and useful information to understand the barriers and motivations for consumers to incorporate insect products in their meals and to help insect growers and the food industry to establish consumer trust and identify possible pathways to larger markets [36].'

Reviewer 3 Report

In the present article, authors provide a scientific opinion about the barriers which characterize the insect sector in the European Union. The overall manuscript is based on the knowledge acquired and collected in the framework on SUSINCHAIN European project. The structure of the manuscript begins with the introduction section, followed by the different topics covered by the project, as well as the impacts achieved. The topic is extremely appropriate to the objectives of this journal and the overall interest in this subject. 

In general, the approach which was applied by the authors is to take into consideration each project contribiutions, by firstly giving an overview about the state of art and thereafter describing the main findings of the project. However, in my opinion, almost all these sections report as background a list of self-citations, which sometimes do not clearly describe the global scenario. The sections about the project contributions should be entirely revised in the initial part and should be written as the paragraph 2.6 (about economic and environmental sustainability). In fact, this was the only one which describes the background without using a list of self-citations, even if authors have already published work about this topic.

In the manuscript are presented some repetition about the different objectives of the project. Please revise the manuscript in order to avoid repetition.

Specific comments:

Please revise the manuscript in order to include abbreviation when already reported in the text.

The manuscript should be reviewed entirely for odd/fragment sentences and grammar.

The title should include a reference to SUSINCHAIN, since the overall manuscript is based on it. As an example: “Overcoming technical and market barriers to enable sustainable large-scale production and consumption of insect proteins in Europe: a SUSINCHAIN perspective”.

Line 81: please substitute “we consider insect species” with “insect species have been proposed”

Line 89: please rephrase

Line 148-148: authors should also underline which barriers have not been addressed by the project

Line 291-298: authors describe the consumer perception about insect consumption, however the citations are referred only to surveys performed before the inclusion of insect in the novel foods category. I suggest authors to revise these references and substitute them with others more recent (just an examples Detilleux et al., 2021; Halonen et al., 2022; Sogari et al. 2019)

Line 486: please include some sentences about the future perspective or key barriers which the project will cover during its last year.

Author Response

Dear reviewer,

With respect to your comments, please see my rebuttal in bold.

In the present article, authors provide a scientific opinion about the barriers which characterize the insect sector in the European Union. The overall manuscript is based on the knowledge acquired and collected in the framework on SUSINCHAIN European project. The structure of the manuscript begins with the introduction section, followed by the different topics covered by the project, as well as the impacts achieved. The topic is extremely appropriate to the objectives of this journal and the overall interest in this subject.

Thanks your for your remark that the topic is extremely appropriate to the objectives of the journal and the interest in the subject.

In general, the approach which was applied by the authors is to take into consideration each project contributions, by firstly giving an overview about the state of art and thereafter describing the main findings of the project. However, in my opinion, almost all these sections report as background a list of self-citations, which sometimes do not clearly describe the global scenario. The sections about the project contributions should be entirely revised in the initial part and should be written as the paragraph 2.6 (about economic and environmental sustainability). In fact, this was the only one which describes the background without using a list of self-citations, even if authors have already published work about this topic.

The approach is indeed to first give an overview about the state of the art and thereafter to describe the main findings of the project. The sections have been revised (except section 2.6) to reduce as much as possible the number of self-citations in the state of the art. However sometimes this could not be avoided since these references include essential information. Of course in parts on the output of the project authors self-citations could not be avoided.

In the manuscript are presented some repetition about the different objectives of the project. Please revise the manuscript in order to avoid repetition.

The manuscript has been revised to avoid repetitions about the objectives in the introduction. In the subsequent sections it cannot be avoided that it is mentioned sometimes that activities refer to the objectives.

Specific comments:

Please revise the manuscript in order to include abbreviation when already reported in the text.

The manuscript has been checked for the use of abbreviations.

The manuscript should be reviewed entirely for odd/fragment sentences and grammar.

The manuscript has been checked for odd/fragment sentences and grammar to the best English language knowledge of the authors.

The title should include a reference to SUSINCHAIN, since the overall manuscript is based on it. As an example: “Overcoming technical and market barriers to enable sustainable large-scale production and consumption of insect proteins in Europe: a SUSINCHAIN perspective”.

Thanks for the suggestion. The title has been changed according to the suggestion.

Line 81: please substitute “we consider insect species” with “insect species have been proposed” - done

Line 89: please rephrase - Changed into: 'require fewer resources to produce the same amount of protein as traditional protein sources' 

Line 148-148: authors should also underline which barriers have not been addressed by the project - The following text is included here: 'As described in the following sub-sections, several of the key barriers are addressed by activities under the SUSINCHAIN project. SUSINCHAIN only addresses research with substrates that are allowed according to European legislation. This means that biowaste sources such as manure, slaughter by-products, and catering waste were not included in the project as substrates.'  

Line 291-298: authors describe the consumer perception about insect consumption, however the citations are referred only to surveys performed before the inclusion of insect in the novel foods category. I suggest authors to revise these references and substitute them with others more recent (just an examples Detilleux et al., 2021; Halonen et al., 2022; Sogari et al. 2019)

The more recent references were included.

Line 486: please include some sentences about the future perspective or key barriers which the project will cover during its last year.

The following sentence has been included to conclude what's the project contributing to and what needs to be done to make the insect chain even more circular: 'The project contributes to these projections for the insect sector and helps to achieve these projections by providing knowledge, data and tools. Insects are classified as 'farmed animals,' and the EU has placed restrictions on the feed that can be supplied to them. Insects intended for human food or animal feed are also subject to these limitations. Only plant-based products are allowed to be fed to insects. However, components of animal origin such as milk, eggs, and their products, honey, rendered fat, and blood products from non-ruminant animals are permitted. It is, however, prohibited to feed farmed animals other slaughtering or rendering-derived products, manure, or catering waste. If the product contains meat or fish, the same prohibition applies to unsold products from supermarkets or food industries (e.g., unsold products due to manufacturing or packaging defects). To completely close the loop and make the insect chain even more sustainable, further information and data on the safety of legally not yet approved substrates to grow insects is required.'

Reviewer 4 Report

This manuscript offers valuable information on the current situation regarding insect production as food and feed. Although insects have attracted considerable interest as a source of nutrients for animal feed and human nutrition, there are still some barriers that retard the progress on their exploitation as a nutrient source. My general impression of this work and its presentation in the form of a manuscript is really positive. To conclude the manuscript is well written and of high quality. Therefore, I highly recommend its publication in "Insects".

Author Response

Dear reviewer,

Thank you for positive impression of this manuscript

Reviewer 5 Report

The paper relates to presenting the main aims and some of the results of SUSINCHAIN, a Horizon2020 funded project, and deals with the main technical and practical barriers that should be overcome to implement the use of insect farming as a sustainable circular economy tool to obtain high value products for feed and food industry. The work is of great interest, considering the practical outcomes of this topic.

Most of the barriers have been addressed, however I would suggest tackling also what is currently probably the major obstacle to the implementation of insect farming in Europe, i.e. the regulation on the feed for the farmed insects. It should be very clear to the readers what are currently the restrictions on the possible substrates that can be used, and the suggestions on how to tackle this issue.

Although the paper is generally well written, it should be more concise, especially in some parts.

Specific comments on the different parts are given below.

L217 sentence incomplete: please check

L238-9 please be more specific here and add references

L270-1 any reference for these results?

L316-362 Please revise, simplify and shorten this paragraph as there are repetitions. Should be more linear to improve readability

L324 please check: I guess that “it selves” should be “itself”;  and include “by” before rearing

L325-326 “…highlighted that cadmium may be of concern in BSF and arsenic in mealworms” possible better this way “…highlighted that cadmium and arsenic may be of concern in BSF and mealworms, respectively”

L331-2 avoid repetition of “such as" 2 times in the same sentence

L334-338 the paragraph need to be simplified

L354-362 also this paragraph can be efficiently condensed

REFERENCES

Please check that all scientific names are correctly written in Italics

Author Response

Dear reviewer,

Thank you for your comments.

Please find below in bold our reply to the comments

The paper relates to presenting the main aims and some of the results of SUSINCHAIN, a Horizon2020 funded project, and deals with the main technical and practical barriers that should be overcome to implement the use of insect farming as a sustainable circular economy tool to obtain high value products for feed and food industry. The work is of great interest, considering the practical outcomes of this topic.

Thank you for underlining the interest of the work.

Most of the barriers have been addressed, however I would suggest tackling also what is currently probably the major obstacle to the implementation of insect farming in Europe, i.e. the regulation on the feed for the farmed insects. It should be very clear to the readers what are currently the restrictions on the possible substrates that can be used, and the suggestions on how to tackle this issue.

At the end of section 3 (the paper concludes with this statement) the following text has been included to stress your comment: 'Insects are classified as 'farmed animals,' and the EU has placed restrictions on the feed that can be supplied to them. Insects intended for human food or animal feed are also subject to these limitations. Only plant-based products are allowed to be fed to insects. However, components of animal origin such as milk, eggs, and their products, honey, rendered fat, and blood products from non-ruminant animals are permitted. It is, however, prohibited to feed farmed animals other slaughtering or rendering-derived products, manure, or catering waste. If the product contains meat or fish, the same prohibition applies to unsold products from supermarkets or food industries (e.g., unsold products due to manufacturing or packaging defects). To completely close the loop and make the insect chain even more sustainable, further information and data on the safety of legally not yet approved substrates to grow insects is required.'

Although the paper is generally well written, it should be more concise, especially in some parts.

The entire document has been checked to make it more concise where possible.

Specific comments on the different parts are given below.

L217 sentence incomplete: please check

Part of the sentence has been changed into: 'are investigated for BSF for the first time [24].'

L238-9 please be more specific here and add references

References are included.

L270-1 any reference for these results?

Results from the project on beneficial health effects are not yet published.

L316-362 Please revise, simplify and shorten this paragraph as there are repetitions. Should be more linear to improve readability

This part in the text has been revised to avoid repetitions.

L324 please check: I guess that “it selves” should be “itself”;  and include “by” before rearing

This has been corrected.

L325-326 “…highlighted that cadmium may be of concern in BSF and arsenic in mealworms” possible better this way “…highlighted that cadmium and arsenic may be of concern in BSF and mealworms, respectively”

The sentence has been changed into: 'Cadmium and arsenic may be of concern in BSF and MW, respectively [42], whereas mycotoxins do not seem to accumulate in these species [43].'

L331-2 avoid repetition of “such as" 2 times in the same sentence

Sentence has been adjusted to avoid the use of 'such as'

L334-338 the paragraph need to be simplified

The paragraph has been simplified

L354-362 also this paragraph can be efficiently condensed

The paragraph has been condensed

REFERENCES

Please check that all scientific names are correctly written in Italics

All scientific names are corrected and written in Italics now.